# Differential Effects of *Furin* Deficiency on Insulin Receptor Processing and Glucose Control in Liver and Pancreatic β Cells of Mice

**DOI:** 10.3390/ijms22126344

**Published:** 2021-06-14

**Authors:** Ilaria Coppola, Bas Brouwers, Sandra Meulemans, Bruno Ramos-Molina, John W. M. Creemers

**Affiliations:** 1Laboratory for Biochemical Neuroendocrinology, Department of Human Genetics, KU Leuven, 3000 Leuven, Belgium; ilaria.coppola@kuleuven.be (I.C.); basbrouwers2@gmail.com (B.B.); sandra.meulemans@kuleuven.be (S.M.); 2Obesity and Metabolism Group, Biomedical Research Institute of Murcia (IMIB-Arrixaca), 30120 Murcia, Spain

**Keywords:** insulin receptor, proprotein convertase, FURIN, liver, pancreatic β cells, insulin signaling, glucose homeostasis

## Abstract

The insulin receptor (IR) is critically involved in maintaining glucose homeostasis. It undergoes proteolytic cleavage by proprotein convertases, which is an essential step for its activation. The importance of the insulin receptor in liver is well established, but its role in pancreatic β cells is still controversial. In this study, we investigated the cleavage of the IR by the proprotein convertase FURIN in β cells and hepatocytes, and the contribution of the IR in pancreatic β cells and liver to glucose homeostasis. β-cell-specific *Furin* knockout (β*Fur*KO) mice were glucose intolerant, but liver-specific *Furin* knockout (L*Fur*KO) mice were normoglycemic. Processing of the IR was blocked in β*Fur*KO cells, but unaffected in L*Fur*KO mice. Most strikingly, glucose homeostasis in β-cell-specific IR knockout (βIRKO) mice was normal in younger mice (up to 20 weeks), and only mildly affected in older mice (24 weeks). In conclusion, FURIN cleaves the IR non-redundantly in β cells, but redundantly in liver. Furthermore, we demonstrated that the IR in β cells plays a limited role in glucose homeostasis.

## 1. Introduction

The IR is a key regulator of glucose homeostasis. The binding of insulin to its receptor in metabolic organs such as liver, pancreas, muscle, brain, and adipose tissue leads to the activation of a signaling pathway mostly aimed at maintaining glucose homeostasis in humans and animals [1].

The IR is synthesized as an inactive precursor protein (proIR) that needs post-translational modifications to be activated. First, it requires endoproteolytic cleavage at the tetrabasic sequence RKRR^752^, yielding disulfide-linked α and β subunits [2]. Patients carrying a mutation in this cleavage site present with severe insulin-resistant diabetes [3,4]. The tetrabasic sequence is most likely removed afterward by carboxypeptidase D, a process that has been shown to be required for full activation of the insulin-like growth factor 1 receptor [5]. The endoproteolytic cleavage step has been shown to be performed by the proprotein convertase (PC) family members FURIN, PACE4, PC5/6 (isoforms A and B), and PC7 in overexpression experiments [6]. However, FURIN was put forward as the physiological PC for processing of the proIR in the secretory pathway and PACE4 at the cell surface when FURIN activity is reduced [6]. In colorectal cell lines and in mouse mammary gland tissue, genetic ablation of *Furin* indeed blocked proIR processing, suggesting non-redundant cleavage by FURIN [7,8]. However, the proIR was normally processed in liver of an inducible *Furin* knockout mouse model, revealing tissue-specific redundancy [9]. Cleavage of the proIR by PCs in pancreatic β cells has not been investigated.

Insulin signaling has been studied extensively in many tissues. In liver, it regulates glucose metabolism by suppressing hepatic glucose production and promoting insulin clearance and glycogen storage [10,11]. Liver-specific IR knockout mice show severe glucose intolerance together with hyperinsulinemia, and subsequent hepatic dysfunction [12]. The physiological consequences of reduced IR signaling in β cells remain controversial. Notably, studies in transgenic mice have suggested an important role of the IR in pancreatic β cells for the regulation of peripheral glucose homeostasis [13,14,15]. However, these studies made use of a Cre driver line shown to have off-target expression in the brain [16], or were affected by the presence of the human growth hormone (hGH) minigene [17,18,19]. Recent studies, using Cre drivers lacking these pitfalls, showed that in β cells of adult islets, the IR controls insulin release and β-cell physiology [20,21].

In this study, we used conditional knockout mouse models and CRISPR-generated knockout cell lines to investigate the impact of *Furin* deficiency on proIR cleavage and insulin signaling in liver and β cells.

## 2. Results

### 2.1. IR Processing and Signaling Are Severely Affected in FurKO β Cells

To determine the role of FURIN in the proteolytic activation of the proIR in pancreatic β cells, we first analyzed the murine β-cell line βTC3. This cell line showed a similar gene expression pattern of PCs active in the constitutive secretory pathway at mRNA levels compared to mouse islets (Figure 1A). *Furin, Pcsk6,* and *Pcsk7*, encoding FURIN, PACE4, and PC7, respectively, were highly expressed, while *Pcsk5*, encoding PC5/6, was not. In βTC3 cells, the lack of *Furin* (*Fur*KO) resulted in severely impaired, if not blocked, proIR cleavage (Figure 1B), and transfection with recombinant *Furin* rescued the cleavage of mature IR (Figure 1C). Importantly, the uncleaved proIR in *Fur*KO β cells was unable to properly respond to insulin based on lack of phosphorylation of IRS1 and AKT after insulin stimulation (Figure 1D,E).

### 2.2. IR Proteolytic Cleavage Is Not Altered in Liver-Specific FurKO Mice

At mRNA level, *Furin* was the most abundant PC in mouse liver, although *Pcsk5* and *Pcsk6* were highly expressed as well (Figure 2A). In vivo, conditional knockout of *Furin* in mouse hepatocytes using the Alb-Cre mice (L*Fur*KO; Figure 3A) induced a modest, non-significant increase of *Pcsk5* and *Pcsk6* gene expression (Figure 2A). We subsequently evaluated the IR processing in hepatocytes from L*Fur*KO and control mice by Western blot. As shown in Figure 2B,C, the cleavage of proIR was not significantly reduced in L*Fur*KO mice compared to the controls, indicating almost complete redundancy for proteolytic cleavage by other PCs.

### 2.3. Impact of Conditional Furin Deletion in Liver and Pancreatic β Cells on Glucose Homeostasis

β-cell-specific *Furin* knockout (β*Fur*KO) mice were generated by breeding *Fur*^fl/fl^ mice [9] with RIP-Cre^+/−^ (Figure 3A). A comparative gene expression analysis of all PCs active in the constitutive secretory pathway, *Furin*, *Pcsk5*, *Pcsk6*, and *Pcsk7* in liver and pancreatic islets of *Fur*^fl/fl^ mice is displayed in Figure 3B and Figure 4. Overall, mRNA expression levels of all tested PCs were significantly higher in mouse liver with respect to isolated mouse islets. We subsequently studied how FURIN deficiency affected glucose homeostasis in vivo in both tissue-specific knockout models. As expected, we observed that L*Fur*KO mice remained glucose tolerant both on chow and a HFD (Figure 3C–F) with normal insulin sensitivity on HFD (Appendix A). In contrast, β*Fur*KO mice were severely glucose intolerant, with significantly higher fasting blood glucose levels on HFD (Figure 3G–H and Appendix A), even on a chow diet, as we described in a previous study [22]. We also did not observe changes in fasting blood glucose and body weight in L*Fur*KO mice compared to controls (Appendix A).

### 2.4. IR Deficiency in Pancreatic β Cells Does Not Induce Severe Glucose Intolerance

IR signaling has distinct roles in liver and pancreas [12,14]. Since the role of the IR in pancreatic β cells is still controversial, we re-evaluated the effect of *Insr* knockout in β cells of mice, using a Cre-driver line without the hGH minigene. We intercrossed RIP-Cre mice [23] with *Insr*-floxed (*Insr*^lox/lox^) animals to generate βIRKO mice (Figure 5A); these mice showed a 68% reduction of *Insr* mRNA in the islets (Figure 5B), consistent with (near) complete inactivation in the β cells. The remaining mRNA expression level was likely linked to the expression of *Insr* in the other cell types of the islets (α, δ cells, and blood vessel endothelial cells), in which the Cre transgene is not expressed. In contrast to earlier observations [13,24], glucose tolerance was not significantly altered in either 12- or 20-week-old mice (Figure 5C–F), and only mildly affected in 24-week-old animals (Figure 5G,H). In addition, fasting blood glucose levels and body weight were unaltered in β*IR*KO mice (Appendix A). Moreover, a previous study reported that *Insr* deficiency in mouse islets and β cell lines led to an induction of ATF4-dependent genes (i.e., Trib3) [25], similar to our previous results using the β*Fur*KO [22]. However, we observed a non-significant reduction in the gene expression levels of *Trib3*, *Chop,* and *Atf4* in isolated islets from the β*IR*KO mice (Figure 5I). This showed that the ATF4 pathway was not altered in these βIRKO mice (Figure 5J).

## 3. Discussion

One of the main goals of this study was to demonstrate that the IR is processed by FURIN in a tissue-specific way, and that the lack of insulin signaling has differential functions in both hepatocytes and pancreatic β cells. Our results revealed that in pancreatic β cells, proIR is non-redundantly cleaved by FURIN. Similar results were observed in colorectal cell lines and mouse mammary gland tissue lacking *Furin* [7,8]. In contrast, in liver of *Fur*KO mice, we show a complete redundancy of proIR cleavage, which is in agreement with our previous work in which an inducible promoter was used to inactivate *Furin* [9]. This tissue specificity in cleaving proIR is likely due to the differential abundance of the other PCs transcripts in the analyzed tissues. For instance, the overall mRNA level of *Pcsk6*, which is the most likely PC to provide redundancy as described in certain cellular models [6], is higher in mouse liver than in mouse pancreatic islets. This supports the hypothesis that the cleavage of a specific substrate in a given tissue can be linked to the availability of compensatory PCs [26]. However, the differential abundance of the PC transcripts in these tissues is only one of the possible explanations of the redundant cleavage of proIR in liver. For instance, the presence of a regulated secretory pathway in β cells but not in hepatocytes likely has an influence on the trafficking of the PCs and proIR. In particular, the process of post-Golgi trafficking through direct constitutive secretion in the liver is distinctly different from the constitutive-like secretion pathway described in β cells [27]. In addition to the tissue-specific enzyme expression, and the cell-specific differences in trafficking of substrates and enzymes, several factors are known to contribute to the enzyme–substrate interaction. The substrate specificity of PCs is overlapping but not identical, and hence is responsible for part of the substrate selectivity. Another crucial factor is the spatiotemporal co-expression of enzyme and substrate. For instance, this is relevant for the precursor of pituitary adenylate cyclase-activating polypeptide (proPACAP), which is cleaved by PC4 in germ cells [28,29], and by PC1/3 and PC2 in neuroendocrine cells [30]. Additional factors that might contribute to tissue-dependent processing are only partly known but can include, for instance, certain post-translational modifications. Thus, glycosylation or phosphorylation near the cleavage site can dynamically affect processing, as was recently shown for FGF23 [31,32]. Overall, this finding of tissue-specific processing adds an extra layer of regulation, allowing the spatiotemporal fine-tuning of IR activity.

The lack of cleavage of the proIR in β*Fur*KO cells results in an impaired response to insulin, indicating that proIR needs to be proteolytically cleaved by FURIN for a functional insulin signal transduction in pancreatic β cells. These results are in line with previous studies in patients and cellular models overexpressing the IR protein with mutations in the cleavage site [3,4,33,34]. Overall, our results highlight an important role of FURIN activity in the regulation of the autocrine functions in β cells through proteolytic activation of IR, as suggested by the alterations in glucose homeostasis observed in β*Fur*KO mice. On the other hand, the redundancy-hypothesis of proIR cleavage in mouse liver was confirmed by the unaffected hepatic IR processing, and normal response to glucose challenge in hepatocyte-specific *Fur*KO mice. Interestingly, these mice showed improved glucose tolerance on NCD, suggesting that inhibition of hepatic *Furin* may protect against glucose dysregulation in metabolic disorders.

Since FURIN has multiple substrates, the glucose intolerance observed in the β*Fur*KO mice cannot automatically be attributed to lack of processing and signaling of the proIR. Therefore, we studied the effect of IR-deficiency in a βIRKO mouse model using the same Cre driver as in the β*Fur*KO mice. Surprisingly, we only observed a mild glucose intolerance in 24-week-old βIRKO mice, and a normal response to a glucose challenge in younger mice. These results are in sharp contrast with a previous study in which βIRKO mice showed a progressive impairment in glucose tolerance [13]. However, for this study, the authors used the RIP-Cre^Mgn^ driver line, which was later shown to exhibit glucose-related phenotype itself [35,36] due to ectopic hGH expression [17]. Using the RIP-Cre^Herr^ driver line, which is not affected by the hGH expression, we could establish that glucose homeostasis is normal in βIRKO mice of similar age, indicating that hGH is the most likely cause of the phenotype of the aforementioned mouse model. Accordingly, in a recent study using an inducible MIP1-Cre knock-in mouse line, inactivation of the IR in adult β cells resulted in normal glucose tolerance, insulin release, and β-cell mass when fed a NCD diet. The mice became glucose intolerant only when challenged with a HFD [20].

Finally, we previously found that FURIN deficiency in pancreatic β cells leads to severe glucose intolerance caused by altered lysosomal acidification and aberrant activation of the mTORC1/ATF4 pathway [22]. Crosstalk between IR signaling and the mTORC1 pathway has been reported in several studies [37,38]. However, we did not observe any effect of the lack of IR in the activation of the mTORC1/ATF4 pathway.

In conclusion, our findings establish a critical role of FURIN in proteolytic processing of the proIR, and regulation of insulin signaling in pancreatic β cells, but not in hepatocytes. We have also established that impaired IR signaling in pancreatic β cells does not result in severe glucose intolerance (Figure 4). These results highlight that processing of substrates by members of the PC family can be entirely tissue-dependent, which should be taken into account for therapeutic strategies.

## 4. Materials and Methods

### 4.1. Mice

Alb-Cre (B6.Cg-Tg(Alb-cre)21Mgn/J) mice, bought from Jackson Laboratories (https://www.jax.org), were bred with *Fur*^fl/fl^ mice to generate hepatocyte-specific *Furin*-deficient mice (L*Fur*KO) [9]. RIP-Cre^+/−^ mice (Tg[Ins2-Cre]23Herr) [23] were bred with animals in which the exon 4 of the *Insr* gene was flanked by loxP sites (*IR*^Lox^) (https://www.jax.org/strain/006955; accessed on 6 September 2016), or with *Fur*^fl/fl^ mice to generate βIRKO and β*Fur*KO mice, respectively. Mice were backcrossed at least 5 times to a C57Bl6J background as previously described [22]. All the mice were housed in standard cages on a 12 h day/night cycle and fed a standard rodent chow or HFD (45 kJ % from fat (lard), 20 kJ % from proteins, and 35 kJ % from carbohydrates; Ssniff, Germany) in a conventional animal facility of the KU Leuven. Food and water were provided ad libitum. To avoid confounding effects of estrogens on glucose homeostasis [39,40], only male mice were included in this study.

### 4.2. Intraperitoneal Glucose and Insulin Tolerance Test (IPGTT and IPITT)

For IPGTT, mice were fasted overnight or 5 h and injected with 2 mg/g body weight (BW) D-glucose (GTT) or 0.75 IU/g BW human insulin (ITT) in PBS, respectively. Blood glucose levels were monitored at indicated time points using a Contour Glucometer (Roche, Basel, Switzerland).

### 4.3. Islet Isolation

Islet were isolated by locally injecting the pancreas with 1 Wünsch unit/mL Liberase (Roche, Basel, Switzerland) in HANKS buffer as previously described [17,22]. Injected pancreata were incubated for 18 min at 37 °C in a shaker (200 rpm), and the islet fraction subsequently was separated from exocrine tissue using a Dextran T70 gradient. Finally, islets were handpicked twice in HANKS buffer under a stereomicroscope to reach a pure islet fraction for further processing.

### 4.4. Cell Culture and Transfection

The mouse insulinoma cell line βTC3 was cultured in DMEM:F12 (1:1) supplemented with 10% heat-inactivated fetal bovine serum. For overexpression experiments, βTC3 cells were transfected with plasmids encoding mouse *Furin* and/or *Insr* (pcDNA3.1 backbone) using Lipofectamine 2000 (Life Technologies, Carlsbad, CA, USA) according to the manufacturer’s protocol. For insulin-stimulation experiments, βTC3 cells were seeded at a cell density of 80%, and the day after they were starved overnight in serum-free DMEM:F12 medium containing 0.1% bovine serum albumin (BSA). Subsequently, cells were cultured for 2 h in glucose-free modified KRB buffer (125 mM NaCl, 4.74 mM KCl, 1 mM CaCl_2_, 1.2 mM KH_2_PO_4_, 1.2 mM, MgSO_4_, 5 mM NaHCO_3_, 25 mM HEPES (pH 7.4)) with 0.1% BSA to minimize endogenous insulin secretion. The stimulation was performed in the same buffer with 100 nM insulin for 5 min.

### 4.5. Generation of Furin Knockout βTC3 Cells Using the CRISPR-Cas9 Nuclease System

The *Furin* knockout βTC3 cell line was generated as previously described [22,41].

### 4.6. Microarray Analysis

The microarray analysis was performed in the previous study, and the microarray data were deposited in the Gene Expression Omnibus (GSE150312) [22].

### 4.7. Crude Membrane Fraction from Mouse Liver

Mouse liver samples were homogenized in 5 volumes of sucrose buffer (sucrose 0.25 M, Tris-HCl 50 mM, MgCl_2_ 5 mM. The post-nuclear supernatant was obtained after centrifugation at 600× *g* at 4 °C for 10 min, and further centrifugation of the supernatant at 10,000× *g* at 4 °C for 30 min. The resulting supernatant was ultra-centrifuged at 25,000× *g* at 4 °C for 1 h (TLA 55 rotor). The pellet was dissolved in 1X cell lysis buffer (Bioke, Leiden, Netherlands) supplemented with complete EDTA-free protease inhibitor cocktails (Sigma, St. Louis, MO, USA) and 2% SDS.

### 4.8. Western Blot

Cells were lysed in 1x RIPA buffer (50 mM Tris-HCl (pH 8.0), 150 mM NaCl, 1.0% NP-40, 0.5% sodium deoxycholate, 0.1% sodium dodecyl sulfate) supplemented with complete EDTA-free protease and a phosphatase inhibitor cocktail (Sigma, St. Louis, MO, USA). The Western blot was performed according to standard procedures. A total of 70 µg of proteins from liver crude membrane fractions and 25 µg from the whole-cell lysate of βTC3 cells were processed for the Western blot. The primary antibodies used were rabbit anti-GAPDH (1:5000, Cell Signaling Technology, Danvers, MA, USA); rabbit anti-IR (insulin Rβ c19, 1:1000, Santa Cruz, Santa Cruz, CA, USA); rabbit anti-phospho-IRS1(Ser318) (1:1000, Cell Signaling Technology); rabbit anti-IRS1 (1:1000, Cell Signaling Technology); rabbit anti-phospho Akt (Ser473) (1:1000, Cell Signaling Technology); and rabbit anti-Akt (1:1000, Cell Signaling Technology).

### 4.9. Quantitative RT-PCR (RT-qPCR)

RNA from snap-frozen mouse liver, freshly isolated mouse islets, or βTC3 cells was extracted using the Nucleospin RNA II (Macherey Nagel, Düren, Germany) kit according to the manufacturer’s protocol. The cDNA was synthetized from 1 µg RNA in 20 µL of reaction volume using the iScript cDNA synthesis kit (Bio-Rad, Hercules, CA, USA). Primers were designed using the ProbeFinder software (Roche Applied Sciences, Basel, Switzerland). The quality of the RNA templates was assessed by using a NanoDrop ND-1000 spectrophotometer (V.3.8.1). RT-qPCR was performed in triplicate with a Bio-Rad CFX96 real-time qPCR detection system (Bio-Rad) using SYBR Green. RT-qPCR was performed with a CFX Connect Real-Time PCR Detection System from Bio-Rad using SYBR Green supermix (Bio-Rad). Data is represented as 2^−ΔCt^. Primers for mouse genes are listed in Appendix A.

### 4.10. Statistical Analysis

Results are expressed as means ± SEM. Statistical analysis was performed by unpaired Student’s *t*-test or one-way ANOVA with Sidak’s multiple comparisons test for grouped analysis, or repeated measure two-way ANOVA for pairwise time-specific differences between genotypes. A value of *p* < 0.05 was considered significant. * *p* < 0.05, ** *p* < 0.01, *** *p* < 0.001.

## Figures and Tables

**Figure 1 ijms-22-06344-f001:**
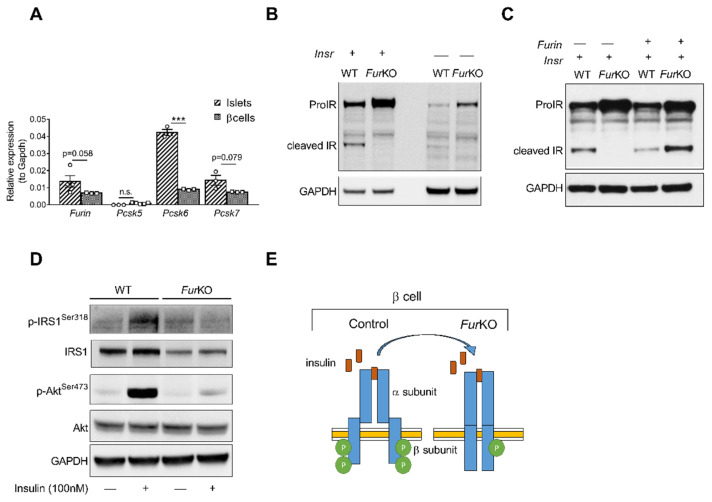
IR processing and signaling is severely affected in FurKO β cells. (**A**) Comparative RT-qPCR analysis of *Furin*, *Pcsk5*, *Pcsk6*, and *Pcsk7* gene expression in the islets of *Fur^fl^*^/fl^ mice (*n* = 3–5 per genotype) and wild-type βTC3 cells. *** *p* < 0.001 determined by two-way ANOVA with Sidak’s multiple comparisons test. All the RT-qPCR data were normalized to *Gapdh*. (**B**) Western blot analysis of the IR protein level in whole-cell lysate of WT and *Fur*KO IR-transfected and non-transfected βTC3 cells. GAPDH was used as loading control. (**C**) Western blot analyses of IR protein levels in whole cell lysates from βTC3 cells transfected with *Insr* or co-transfected with *Insr* and *Furin*. GAPDH was used as loading control. (**D**) Western blot analysis of phosphorylated and total IRS1, and Akt in whole cell lysates from βTC3 cells treated with 100 nM insulin or vehicle for 5 min. GAPDH was used as loading control. (**E**) Schematic representation of a suggested mechanism of IR signaling in *Fur*KO βTC3 cells compared to a physiological condition.

**Figure 2 ijms-22-06344-f002:**
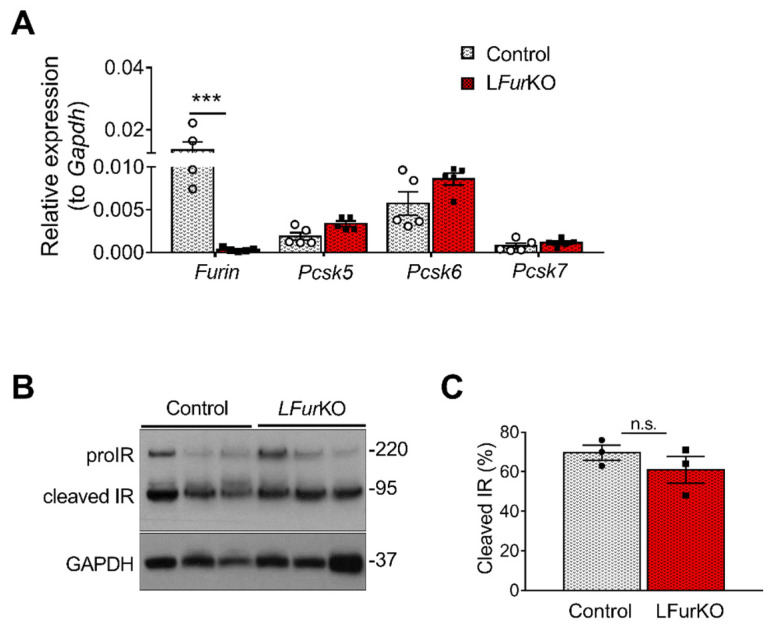
IR proteolytic cleavage is not altered in the liver of *Furin* knockout mice. (**A**) RT-qPCR analysis of *Furin*, *Pcsk5*, *Pcsk6*, and *Pcsk7* gene expression in the liver of either L*Fur*KO or control (*Fur*^fl/fl^) mice, *n* = 5 per genotype. *** *p* < 0.001 determined by two-way ANOVA with Sidak’s multiple comparisons test. All RT-qPCR data were normalized to *Gapdh*. (**B**) Western blot analysis of IR processing from liver membrane fractions of control and L*Fur*KO mice, *n* = 3 animals per genotype. GAPDH was used as loading control. (**C**) Protein-level quantification reported as % of cleaved IR over the total IR (ProIR + cleaved IR). ProIR data are represented as the mean ± SEM. Non-significant differences were detected by unpaired *t*-test.

**Figure 3 ijms-22-06344-f003:**
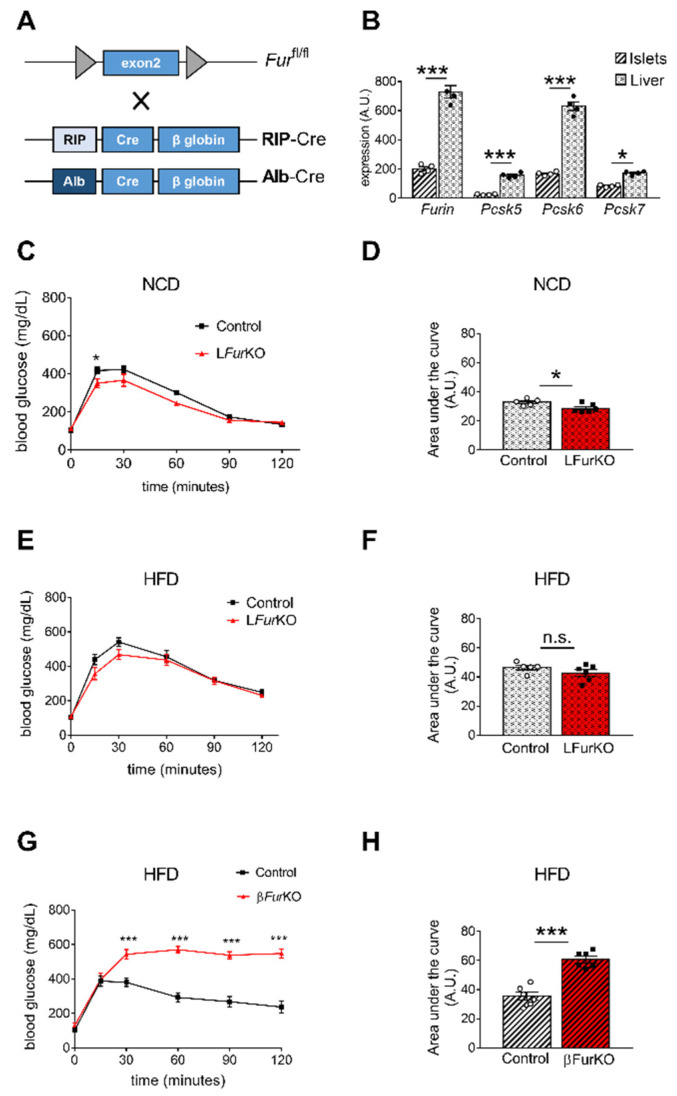
Impact of *Furin* knockout in liver and pancreatic β cells on glucose homeostasis (**A**) Schematic representation of the breeding approach. Mice in which the exon 2 of *Furin* had been floxed (*Fur*^fl/fl^) were crossed with RIP-Cre or Alb-Cre driver lines. (**B**) Microarray analysis of *Furin*, *Pcsk5*, *Pcsk6*, and *Pcsk7* gene expression in liver versus islets of control (*Fur*^fl/fl^) mice, *n* = 4 per genotype. * *p* < 0.05; *** *p* < 0.001 determined by two-way ANOVA with Sidak’s multiple comparisons test. All the data were normalized to *Gapdh.* (**C**) Intraperitoneal glucose tolerance tests (IPGTTs) on 8-week-old male L*Fur*KO and control mice on normal chow diet (NCD) (*n* = 5–6 mice/group). * *p* < 0.05 determined by two-way ANOVA with Sidak’s multiple comparisons test. (**D**) The area under the curve was expressed as g/dL/120 min; *n* = 5–6 mice/group. * *p* < 0.05 determined by unpaired *t*-test. (**E**) Intraperitoneal glucose tolerance test (IPGTT) on 16-week-old male L*Fur*KO and control (*Fur*^fl/fl^) mice fed for 8 weeks on a high-fat diet (HFD, 45% kJ from fat). No significant differences were found by two-way ANOVA with Sidak’s multiple comparisons test. (**F**) The area under the curve was expressed as g/dL/120 min; *n* = 5–6 mice/group. No significant differences were observed by unpaired *t*-test. (**G**) IPGTT on 16-week-old male β*Fur*KO and control (*Fur*^fl/fl^) mice fed for 8 weeks on a HFD (45% kJ from fat). *** *p* < 0.001 determined by two-way ANOVA with Sidak’s multiple comparisons test. (**H**) The area under the curve was expressed as g/dL/120 min; *n* = 6 mice/group. *** *p* < 0.001 determined by unpaired *t*-test. All data were presented as mean ± SEM.

**Figure 4 ijms-22-06344-f004:**
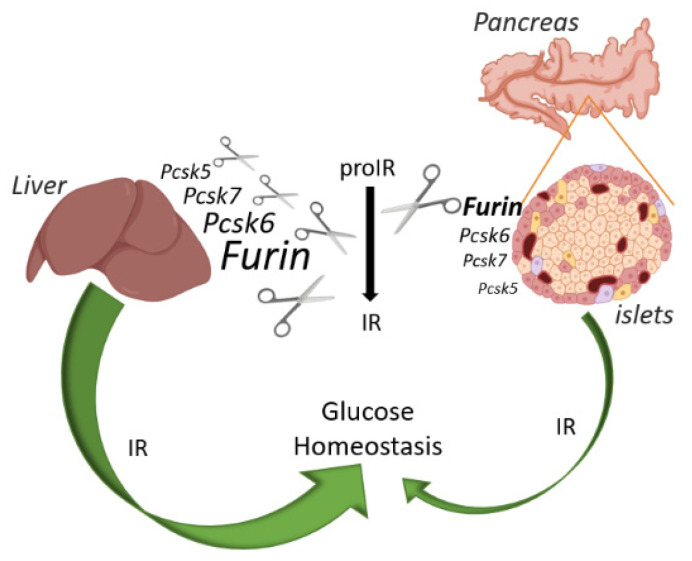
Schematic overview of the IR action on glucose and relative abundance of each PC in liver and pancreatic islets of mice. FURIN non-redundantly cleaves the proIR in pancreatic beta cells of mice, but the IR exerts only a minor effect on glucose homeostasis. In contrast, proprotein convertases other than FURIN can process proIR in the mouse liver, but here this receptor plays a pivotal role in glucose homeostasis. Image created with BioRender.com.

**Figure 5 ijms-22-06344-f005:**
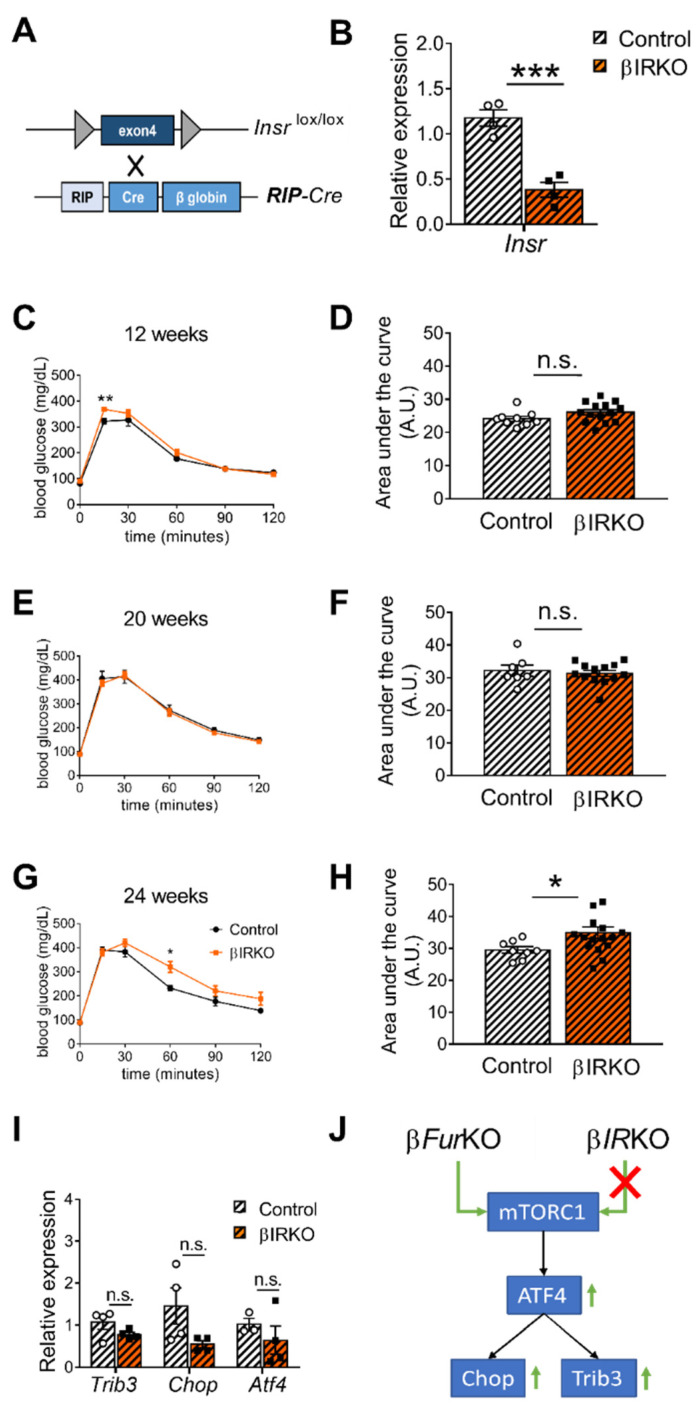
*Insr*-knockout in pancreatic β cells (βIRKO) mildly impairs glucose tolerance in older mice. (**A**) Schematic representation of the breeding approach. Mice expressing the *Insr* gene in which the exon 4 had been floxed (*Insr*^lox/lox^) were crossed with the RIP-Cre driver line. (**B**) RT-qPCR analysis of *Insr* in the islets of control (*Insr*^lox/lox^) and βIRKO mice, *n* = 4 mice/group. All data were normalized to *Gapdh* and to the controls. *** *p* < 0.001 determined by unpaired *t*-test. (**C–H**) IPGTT on male βIRKO and control mice of 12 weeks (*n* = 9–14 mice/group) (**C**), 20 weeks (*n* = 7–13 mice/group) (**E**), and 24 weeks (*n*= 8–16 mice/group) (**G**). Quantification of the area under the curve for (**C**,**E**,**G**) was expressed as g/dL/120 min, and plotted in (**D**,**F**,**H**). * *p* < 0.05, ** *p* < 0.01 determined by two-way ANOVA with a Sidak’s multiple comparisons test. * *p* < 0.05 determined by unpaired *t*-test. All data were presented as mean ± SEM. (**I**) RT-qPCR analysis of *Trib3*, *Chop,* and *Atf4* gene expression in the islets of control (*Insr*^lox/lox^) and βIRKO mice, *n* = 3–4 mice/group. No significant differences were detected by two-way ANOVA with Sidak’s multiple comparisons test. All the RT-qPCR data were normalized to the *Gapdh*. (**J**) Schematic of mTORC1/ATF4 pathway, which was upregulated in β*Fur*KO but not in βIRKO mice.

## Data Availability

The microarray data were deposited in the Gene Expression Omnibus (GSE150312) [22].

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
