# Peer review of "Differential Effects of *Furin* Deficiency on Insulin Receptor Processing and Glucose Control in Liver and Pancreatic β Cells of Mice"

_ijms, 2021, doi:10.3390/ijms22126344_

Round 1

Reviewer 1 Report

The manuscript is written in a precise, logic way. The language is a good quality for non-native speaker.

The authors revealed that in pancreatic β cells, proIR is non-redundantly cleaved by FURIN. In contrast, in liver of FurKO mice, a complete redundancy of proIR cleavage was found. This tissue specificity is likely due to the differential abundance of the other PCs transcripts. The overall mRNA level of Pcsk6, is higher in mouse liver than in mouse pancreatic islets. The lack of cleavage of the proIR in βFurKO cells resulted in an impaired response to insulin, indicating that proIR needs to be proteolytically cleaved by FURIN for a functional insulin signal transduction in pancreatic β cells. The redundancy-hypothesis of proIR cleavage in mouse  liver was confirmed by the unaffected hepatic IR processing and normal response to glucose challenge in hepatocyte-specific FurKO mice. The authors studied the effect of IR-deficiency in a βIRKO mouse model using the same Cre driver as in the βFurKO mice. They observed a mild glucose intolerance in 24-week-old βIRKO mice and a normal response to a glucose challenge in younger mice, what was in sharp contrast with a previous study. The explanation was use of the RIP-CreMgn driver line in previous study, which was later shown to exhibit glucose-related phenotype itself due to ectopic hGH expression. The authors did not observed any effect of the lack of IR in the activation of the mTORC1/ATF4 pathway.  In conclusion, authors suggested a critical role of FURIN in proteolytic processing of the IR, and regulation of insulin signaling in pancreatic β cells, but not in hepatocytes. Impaired IR signaling in pancreatic β cells did not result in severe glucose intolerance.

Minor revision

The manuscript lack of explanation what is the background of difference between pancreatic β cells and liver. Is it only the difference in the overall mRNA level of Pcsk6? What can be the role of the difference in physiology and pathology?  The authors stress that processing of substrates by members of the PC family can be entirely tissue-dependent. Taking into consideration the authors experience in the field a proposal of regulation pathway should be presented.

Author Response

The manuscript is written in a precise, logic way. The language is a good quality for non-native speaker.

The authors revealed that in pancreatic β cells, proIR is non-redundantly cleaved by FURIN. In contrast, in liver of FurKO mice, a complete redundancy of proIR cleavage was found. This tissue specificity is likely due to the differential abundance of the other PCs transcripts. The overall mRNA level of Pcsk6, is higher in mouse liver than in mouse pancreatic islets. The lack of cleavage of the proIR in βFurKO cells resulted in an impaired response to insulin, indicating that proIR needs to be proteolytically cleaved by FURIN for a functional insulin signal transduction in pancreatic β cells. The redundancy-hypothesis of proIR cleavage in mouse liver was confirmed by the unaffected hepatic IR processing and normal response to glucose challenge in hepatocyte-specific FurKO mice. The authors studied the effect of IR-deficiency in a βIRKO mouse model using the same Cre driver as in the βFurKO mice. They observed a mild glucose intolerance in 24-week-old βIRKO mice and a normal response to a glucose challenge in younger mice, what was in sharp contrast with a previous study. The explanation was use of the RIP-CreMgn driver line in previous study, which was later shown to exhibit glucose-related phenotype itself due to ectopic hGH expression. The authors did not observe any effect of the lack of IR in the activation of the mTORC1/ATF4 pathway.  In conclusion, authors suggested a critical role of FURIN in proteolytic processing of the IR, and regulation of insulin signaling in pancreatic β cells, but not in hepatocytes. Impaired IR signaling in pancreatic β cells did not result in severe glucose intolerance.

We would like to thank the reviewer for his/her positive comments.

Comment 1: The manuscript lack of explanation what is the background of difference between pancreatic β cells and liver. Is it only the difference in the overall mRNA level of Pcsk6?  

Response to comment 1: The reviewer raises a very important issue. It is clear that the explanation is more complex than just expression levels of enzymes and substrates. We already discuss (lines 41-43) that the two splice variants of the IR are trafficked slightly different to the plasma membrane, affecting the differential processing by FURIN and PACE4. In addition, the presence of a regulated secretory pathway in β cells but not in hepatocytes is likely to have an influence on trafficking of the enzymes and substrate [1,2]. In particular, the process of post-Golgi trafficking through direct constitutive secretion is distinctly different from constitutive-like secretion. Although the factors influencing cell-specific enzyme-substrate relationships are incompletely understood and certainly underappreciated and often even ignored, we agree with the reviewer that this needs to be discussed. We have now adapted the text as follows:

However, the differential abundance of the PC transcripts in these tissues is only one of the possible explanations of the redundant cleavage of proIR in liver. For instance, the presence of a regulated secretory pathway in β cells but not in hepatocytes likely has an influence on the trafficking of the PCs and proIR. In particular, the process of post-Golgi trafficking through direct constitutive secretion in the liver is distinctly different from the constitutive-like secretion pathway described in β cells [27]” (lines 188-194, page 7).

Comment 2: What can be the role of the difference in physiology and pathology?  

Response to comment 2: This question is interesting, but we can only speculate. The timely processing of substrates adds an extra layer of regulation, allowing the spatiotemporal fine-tuning of activity. We have included this argument in the text

Overall, this finding of tissue-specific processing adds an extra layer of regulation, allowing the spatiotemporal fine-tuning of IR activity” (lines 204-205, page 7).

Comment 3: The authors stress that processing of substrates by members of the PC family can be entirely tissue-dependent. Taking into consideration the authors experience in the field a proposal of regulation pathway should be presented.

Response to comment 3. Although enzyme-substrate relations are not fully understood, there are several factors that are known to contribute. The enzyme needs to be able to cleave the substrate. The substrate specificity of PCs is overlapping but not identical and hence is responsible for part of the substrate selectivity. Another crucial factor is the spatiotemporal co-expression of enzyme and substrate. Additional factors that might contribute to tissue dependent processing are only partly known but can e.g. be cell-specific differences in trafficking or other posttranslational modifications. We have added this information, including some examples, to the discussion:

“In addition to the tissue-specific enzyme expression, and the cell-specific differences in trafficking of substrates and enzymes, several factors are known to contribute to the enzyme-substrate interaction. The substrate specificity of PCs is overlapping but not identical, and hence is responsible for part of the substrate selectivity. Another crucial factor is the spatiotemporal co-expression of enzyme and substrate. For instance, this is relevant for the precursor of pituitary adenylate cyclase-activating polypeptide (proPACAP), which is cleaved by PC4 in germ cells [28,29] and by PC1/3 and PC2 in neuroendocrine cells [30]. Additional factors that might contribute to tissue-dependent processing are only partly known but can include, for instance, certain post-translational modifications. Thus, glycosylation or phosphorylation near the cleavage site can dynamically affect processing, as was recently shown for FGF23 [31,32]” (lines 194-204, page 7).

Reviewer 2 Report

The Coppola et al.’s Ms addresses the effects of tissue-specific knock-outs of furin in the liver and pancreas and their involvement in insulin receptor processing. The paper is original; their observed results are novel and deserve to be published. However, there are some aspects that require further elaboration.

Major points.

  1. The Ms should be adapted to MIQE guidelines (Clin Chem. 2009, 55(4):611-22) and Table S1 must report accession of the sequence, length of amplicons, locations of primers, primer concentrations, and efficiency of amplification.
  2. Dietary fat is poorly described. Which was the composition of macronutrients? What kind of fat was used?
  3. Why were the experiments only carried out in male mice?
  4. In Figure 3 C and D, the hepatic-specific knock-out showed lower levels of glucose on chow diet than control mice. However, the insulin challenge was not influenced, nor the cleaved IR. This aspect should be discussed

Minor points

  1. The title should include the animal model
  2. Gapdh should be in italics in legend of Figure 1
  3. According to MIQE, the procedure should be renamed as RT-qPCR
  4. In Western description of Methods, units are without space when referred to μg

Author Response

The Coppola et al.’s Ms addresses the effects of tissue-specific knock-outs of furin in the liver and pancreas and their involvement in insulin receptor processing. The paper is original; their observed results are novel and deserve to be published. However, there are some aspects that require further elaboration.

We would like to thank the reviewer for his/her positive comments.

Major points:

Comment 1: The Ms should be adapted to MIQE guidelines (Clin Chem. 2009, 55(4):611-22) and Table S1 must report accession of the sequence, length of amplicons, locations of primers, primer concentrations, and efficiency of amplification.

Response to comment 1: Table S1 was accordingly adapted, and the M&M section was modified as follows:

  • Line 323 was changed from “RNA from mouse islets or βTC3 cell was isolated” to “RNA from snap frozen mouse liver, freshly isolated mouse islets or βTC3 cells was extracted”
  • Lines 325-326 were changed from “cDNA was synthetized using the iScript cDNA synthesis kit (Bio-Rad).” to “cDNA was synthetized from 1µg of RNA in 20 µl of reaction volume using the iScript cDNA synthesis kit (Bio-Rad)”
  • Lines 327-328, the following sentence was added: “The quality of the RNA templates was assessed by using a NanoDrop ND-1000 spectrophotometer (V.3.8.1)”.

Comment 2: Dietary fat is poorly described. Which was the composition of macronutrients? What kind of fat was used?

Response to comment 2: We thank the reviewer for indicating the missing information. We adapted the text as suggested:

  • Lines 261-262 were changed from “All the mice were housed in standard cages on a 12-hour day/night cycle and fed a standard rodent chow or high fat diet (HFD, 45% KJ from fat)” to “All the mice were housed in standard cages on a 12-hour day/night cycle and fed a standard rodent chow or high fat diet (HFD, 45 kJ % from fat (lard), 20 kJ % from proteins, and 35 kJ % from carbohydrates; Ssniff, Germany)”

Comment 3: Why were the experiments only carried out in male mice?

Response to comment 3: We thank the reviewer for noticing this missing piece of information. We have now added the following paragraph in the manuscript to clarify this point:

  • Lines 263-265: “To avoid confounding effects of estrogens on glucose homeostasis [40,41], only male mice were included in this study.”

Comment 4: In Figure 3 C and D, the hepatic-specific knock-out showed lower levels of glucose on chow diet than control mice. However, the insulin challenge was not influenced, nor the cleaved IR. This aspect should be discussed

Response to comment 4: We appreciate the reviewer’s comment. As suggested we have discussed this issue in the new version of the manuscript:

  • Lines 215-217: “Interestingly, these mice showed improved glucose tolerance on NCD, suggesting that inhibition of hepatic Furin may protect against glucose dysregulation in metabolic disorders.

Minor points:

Comment 5: The title should include the animal model

Response to comment 5: Thanks for the suggestion. We have now adapted the title as follows: Differential effects of Furin deficiency on insulin receptor processing and glucose control in liver and pancreatic β cells in mice”.

Comment 6: Gapdh should be in italics in legend of Figure 1

Response to comment 6: Thank you for pointing it out. We now changed “Gapdh” to “Gapdh” in Figures 1, 2 and 3.

Comment 7: According to MIQE, the procedure should be renamed as RT-qPCR

Response to comment 7: Thank you for comment. We adapted the manuscript as suggested.

Comment 8: In Western description of Methods, units are without space when referred to μg

Response to comment 8: We have now changed Methods section as suggested by the reviewer.